# Unmet need for treatment-seeking from public health facilities in India: An analysis of sociodemographic, regional and disease-wise variations

**Rajaram Yadav**[1], **Jeetendra Yadav**[2]*, **Chander Shekhar**[3]

**1** ICMR-Regional Medical Research Center, Gorakhpur, India, **2** ICMR-National Institute of Medical Statistics, Ansari Nagar, New Delhi, India, **3** Department of Fertility and Social Demography, International Institute for Population Sciences (IIPS), BSD Marg, Mumbai, Maharashtra, India

* jeetu.nims@gmail.com

**Data Availability Statement:** The present article is based on three rounds of the National Sample Survey (2004, 2014, 2018) data which is freely

## Abstract

Treatment-seeking behaviour is closely associated with the health status of individuals and countries. About 800 million people have no access to health services in the developing world. Though the situation has been improving, the inequalities across geographical regions, socioeconomic status, and disease types continued to persist. The available literature suggests research gaps in examining the unmet need for treatment-seeking from public health facilities across sociodemographic characteristics, regions, and specific diseases. Data for this study comes from the three rounds of National Sample Survey (NSS) (2004, 2014, 2018). We applied descriptive, bivariate, and multivariable analysis to investigate the unmet need for treatment-seeking for public health facilities across sociodemographic characteristics, regions, and specific diseases between 2004 and 2018. The unmet need for treatment-seeking from public health facilities remained high at 60% in 2004 to 62% in 2018. However, the proportion of respondents who did not seek treatment has reduced 12% to 3% from 2004 to 2018. In states like Andhra Pradesh, Madhya Pradesh, Maharashtra, Punjab, Telangana, Uttar Pradesh, and West Bengal, the unmet need for treatment-seeking from public health facilities was more than 60% in 2018. For 2018, the quality of services at public health facilities was the main reason for showing a higher unmet need for treatment-seeking in the richer MPCE quintiles. On the other hand, the ailment not considered serious as the main reason for the unmet need for treatment-seeking from any sources has got nearly doubled from 36% in 2004 to 71% in 2018. This study concludes that improving the availability of various kinds of services at public health facilities should be a priority under India's universal health coverage program. Education plays a vital role in treatment-seeking. Thus, there is an urgent need for increasing awareness among people for treatment-seeking. Ensuring a minimum quality of health care services and reducing long waiting timing would reduce the apathy to receive services from the public health facilities.

available to individuals on https://www.mospi.gov.in/download-tables-data.

**Funding:** The authors received no specific funding for this work.

**Competing interests:** The authors have declared that no competing interests exist.

## Introduction

In the developing world, about 800 million people have no access to health services [1]. Health services have always been poor and inaccessible to a large portion of the population in India, a fact which has also been acknowledged by the government [2,3]. In all the developing countries, except India and Pakistan, the utilization of public facilities is more than that of private facilities for inpatient services. There is high utilization of public services in countries with health expenditure encompassing a larger share of total government expenditure. This is because the poorest quintile is more likely to use public services than the richest quintile in most countries [4]. The reduction of inequality in health and healthcare utilization is the main focus of any healthcare system [5]. Among the prominent causes of low utilization of public health care services are absenteeism of health workers, poor facilities, and low satisfactory level of care [6,7]. This tendency has resulted in decreased faith and, consequently, a rapid increase in the use of private health care services [8]. According to an estimate from NSS data from 1998, the private sector accounted for four-fifths of outpatient care and nearly half of inpatient care.

The Planning Commission of India constituted a high-level expert group on universal health coverage (UHC) in October 2010. The work assigned to the group was to develop a framework for providing affordable and readily available healthcare to all Indians. Universal health coverage was defined as providing equitable health services to all Indian citizens living in any part of the country regardless of gender, caste, religion, social group, or income level. The need for health services is not uniform across the states and far exceeds the current level of provision. In most primary health centers, the availability of human resources and essential requirements, such as hospital beds, physical infrastructure, drugs, and diagnostics, is far below the prescribed norms [9]. Though access to health care has been improving, inequalities across geography, socioeconomic status, and gender continue to persist [10]. The shortage of health workers in the country and inequality in their distribution across states have resulted in large regional variation in access to services [11]. There has been a rapid increase in private healthcare facilities in the urban areas, which created an unequal urban-rural access in healthcare [12]. Also, people residing in places having electricity, high social security and high literacy have a high probability of using formal medical care [13].

The demand for health care depends on many factors like religion, caste, culture, social status, gender, and quality of services [14]. The National Family Health Survey (International Institute for Population Sciences [15] suggests huge socioeconomic and regional inequalities in the health outcomes, with the poor, the lower castes, and the less developed states showing a disproportionally high under-five mortality. And the use of private facilities is more among the general castes than the other castes [16]. Inequality in the availability and utilization of services among different socioeconomic groups and states may be responsible, to a certain extent, for the above-said conditions. The interstate variation in the utilization of services may be best shown by comparing Uttar Pradesh with Kerala. The former is among the worst states, while the latter is the best state regarding health care utilization [17]. Compared with the other states of the country, Kerala has the highest rate of outpatient care, with 88% percent of ill people reported using outpatient care services [18]. The utilization of healthcare services is lower among the tribal populations [19], who depend heavily on public healthcare services [20]. The factors which hinder them from using the services are their geographic location, traditional beliefs, poor economic status, and unavailability of services [19]. The utilization of services is also low in the northeastern states due to the difficult terrain characterized by frequent floods, landslides, and remote villages, which raise barriers in accessing the facilities [21].

### A brief description of India's public healthcare system functioning

Here, it is necessary to highlight briefly how the public health system works in India. The public health facilities in India do not provide "medical consultation" alone as per the protocol. So, either they treat or refer a patient to a higher level of facilities if the treatment could not be given for any reason. In rare circumstances/in an emergency when patients could neither be treated at a particular facility nor referred to a higher-level facility, they are provided medical prescription/consultation for seeking treatment from other sources, including private and others (profit or non-profit) facilities.

The public health system is a three-tier health system. At the primary care level, minor ailments and maternal and child health-related services are offered at the sub-health centre (SHC) and primary health centre (PHC). Primary healthcare denotes the first level of contact between individuals and the health system. It provides care that includes family planning, immunization, prevention of locally endemic diseases, treatment of common diseases or injuries, provision of essential facilities, health education, provision of food, nutrition and sanitation. Secondary healthcare refers to the second tier of the health system where the referral cases from the primary healthcare level facilities receive treatment by the specialists in a higher-level hospital setting. The secondary level facilities include the district hospital (DH) at the district level and the Community Health Centre (CHC) at the block level. Finally, the tertiary healthcare level refers to the third tier of the health system, in which specialized consultative and intensive care is provided, usually on referral from primary and secondary level facilities. The third level healthcare facilities include medical colleges, trauma centres, large specialized district hospitals or medical research Institute's hospitals.

### Rationale of the study

The country has made considerable progress in the health infrastructure since the National Rural Health Mission (NRHM) was launched. However, the progress remained uneven across the regions, with large-scale inter-state variations [11,17,22]. The accessibility to health care facilities is extremely limited in many backward regions of the country. While around 67% population of India lives in rural areas, only 20% of hospital beds are allocated to them [23].

This paper aims to fill the research gaps in examining the unmet need for treatment-seeking, especially for public health facilities across sociodemographic characteristics, regions, and specific diseases between 2004 and 2018. The idea of taking three data points spread over nearly one and a half decade was to see whether different policy initiatives and programs undertaken by the national government has improved the situation of treatment-seeking from public sources or not. A few flagship programs launched during these years at the national level include the National Rural Health Mission in 2005, the Universal Health Coverage (UHC) in 2012, the National Health Mission (NHM) in 2013 and the National Health Policy (NHP) in 2017. In fact, one of the objectives of carrying the NSS on social consumption is to periodically assess the progress of the current interventions and future policy implications. Several studies have also made such attempts in the past [24–26].

### Data and methods

### Ethics statement

The present study used the data from three rounds of the National Sample Survey (2004, 2014, 2018). First, the NSS obtained ethical consent from the review committee of NSS before the survey. Second, a written consent was obtained once the respondent agreed to participate in the data collection during the survey. Therefore, no ethical approval is required separately for the present study.

## Data sources, research design, and sample size

The present study used secondary cross-sectional datasets from NSS 2004 (60[th] round), 2014 (71[th] round), and 2018 (75[th] round). In India, NSS is a nationally-representative large-scale survey conducted under the stewardship of the Ministry of Statistics and Programme Implementation (MoSPI), Government of India [27–29]. NSS uses a two-stage stratified sampling to select households from the urban and rural areas. In the first stage of sampling, villages/urban blocks are selected using the probability proportional to size (PPS) sampling, while in the second stage, 22 households are selected using systematic random sampling from the selected villages/urban blocks. The use of a multistage sampling design promises that the probability sampling is, theoretically, representative of India. The 60[th] round of NSS, held during January-June 2004, covered 383,338 individuals (250,775 rural and 132,563 urban), the 71[st] round, held during January-June 2014, covered 333,104 individuals (189,573 rural and 143,531 urban), and the 75[th] round, held during July 2017-June 2018, covered 5,55,115 individuals (3,25,883 rural and 2,29,232 urban). This study utilised data on 38,753 individuals from the 60[th] round, 37,282 individuals from the 71[st] round, and 43,239 individuals from the 75[th] round who had reported any morbidity during 15 days prior to the survey date. The surveys covered all the states and union territories of India. S1 Table provides the list of the disease conditions used in the study samples [27–30].

## Variables

**Outcome variables.**   The present study used four outcome variables, namely unmet need for treatment-seeking from any sources, the main reason for not seeking treatment at all from any sources, unmet need for treatment-seeking from any public health facilities, and the main reason for unmet need for treatment-seeking from any public health facilities. This study explored detailed analysis by using the unmet need for treatment-seeking from any public health facility and the main reason for the unmet need for treatment-seeking from public health facilities as an outcome variable in the binary logistic regression and multinomial logistic regression, respectively.

The operational definitions of all the outcome variables are given below.

**Unmet need for treatment-seeking from any sources.**   Unmet need for treatment-seeking was determined in terms of the proportion of outpatients those who did not seek treatment at all either from any private or public health facility among those outpatients who suffered from any disease during 15 days prior to the date of survey [27–29,31].

**Reason for not seeking treatment from any sources.**   NSS data provides an opportunity to tabulate the main reason an outpatient could not seek treatment from any sources. We categorized these reasons into five major categories-no facility available in the neighbourhood, financial problem, long waiting time, ailment not considered serious, and others [27–29].

**Unmet need for treatment-seeking from public health facilities.**   The unmet need for treatment-seeking from public health facilities was determined in terms of the proportion of outpatients those who did not seek treatment from any public health facility among those outpatients who suffered from any disease during 15 days prior to the date of survey [27–29].

**Reasons for unmet need for treatment-seeking any public health facilities.**   NSS data the main reason due to which an outpatient could not seek treatment from any public health facilities and these include: required specific services not available, quality satisfactory but facility too far, services available but quality not satisfactory/doctor not available, quality satisfactory but long waiting time, financial constraints, preference for a trusted doctor/hospital, and others. On the basis of frequency and nature of reason, we have categorized them into five major groups-required service not available at public health facility, quality, waiting, financial and others [27–29].

## Socioeconomic explanatory variables

The present study included a number of individual, household and community level variables to examine socioeconomic disadvantages associated with the unmet need for treatment-seeking among those who suffered from any disease in 15 days prior to the survey date. The inclusion of each level variable was based on the theoretical and observed status as applied in the literature and the information available in the NSS dataset in all three rounds of the survey [6,14,19,27–29]. The description of each explanatory variable and the measurement scales are given in S2 Table.

## Statistical analysis

We applied descriptive, bivariate, and multivariable statistical analyses to assess the level of unmet need for treatment-seeking. Descriptive analysis was carried out to identify the sociodemographic characteristics of the patients, and bivariate analysis was applied to dig out the sociodemographic differences in unmet need for treatment-seeking behaviour. We have also used Chi-square test statistics to know the association between unmet need for treatment-seeking behaviour and sociodemographic characteristics.

Multivariable logistic regression is used when the dependent variable is dichotomous, and the independent variables are either categorical or continuous. The equation for the logistic regression is as follows:

$$\text{Logit } (Y) = ln\left(\frac{p}{1-p}\right) = \alpha + \beta_1 x_1 + \beta_2 x_2 + \ldots\ldots\ldots\ldots + \beta_n x_n + \in$$

Where, p is the probability of an outpatient having an unmet need for treatment-seeking from any public health facility, Y is the odds of being in unmet need against not in an unmet need, Xi's are independent variables, and βi's are coefficients. $\alpha$ and $\epsilon$ are intercept and error terms in the model. The dependent variable was dichotomous and coded as '1s' and '0s'; those outpatients who did not seek treatment from any public health facility coded as '1', otherwise '0'.

Finally, a multinomial logistic regression (MLR) was conducted to examine an independent association between the respondent's sociodemographic characteristics and the reasons for not seeking treatment from a public health facility. Such a regression is used in those cases where the dependent variable contains responses in more than two categories, and the independent variables are either categorical or continuous. The dependent variable in our model is the main reason for the unmet need for treatment-seeking from any public health facilities. The main reason codes were '0' for the availability of any public health facility in the neighbourhood, '1' for quality of services, '2' for waiting time, '3' for the financial problem and '4' for others. The model produced the adjusted relative risk ratio of the probability of not seeking treatment from any public health facility for a given reason with respect to the probability of not seeking treatment from any public health facility due to the unavailability of required services. The model is given in the form of a generalized equation as follows:

$$\ln\left(\frac{p_i}{p_1}\right) = \alpha_i + \beta_{ij} x_{ij} + \in_i$$

Where, $p_1$ is the probability of the unmet need for treatment-seeking due to the unavailability of required services, $p_i$ is the corresponding probabilities of unmet need for treatment-seeking for the main reason coded as i = 2,3, and 4. $X_{ij's}$ and $\beta_{ij's}$ are corresponding covariates and coefficients, respectively. $\alpha_i$ and $\epsilon_i$ are intercept and error terms, respectively for i = 2,3, and 4 in the model [32].

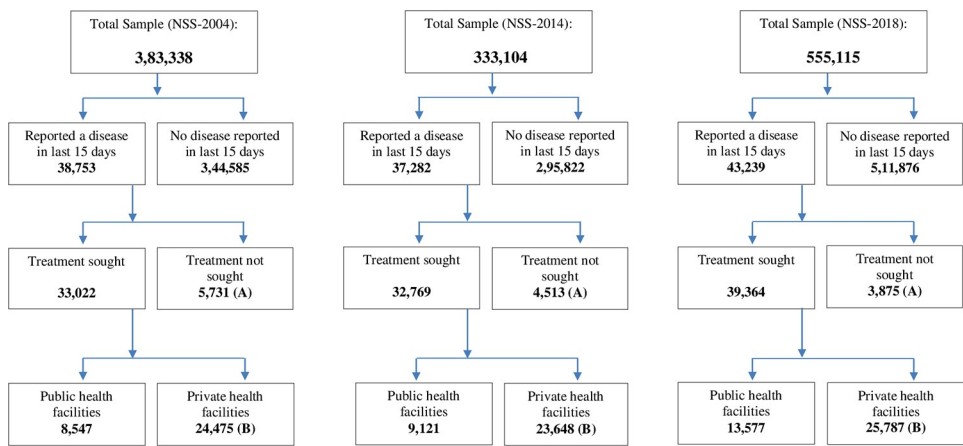

Note: The respondents belong to the group of (A) and (B) in each round of the survey have been considered in the unmet need for seeking treatment from public health facilities.

**Fig 1. Analytical total sample at the Individual level in NSS-2004, 2014 and 2018, India.**

Before inserting the explanatory socioeconomic variables in the multinomial model, we screened the variables for multicollinearity for all three NSS rounds. Multicollinearity arises when the predictors in the model are highly correlated, that is, when the value of VIF>5 [33]. No issue of collinearity among the explanatory socioeconomic variables were detected. The highest variance inflation factor (VIF) was 1.78 for the 60th round, 1.43 for the 71st round, and 2.74 for the 75th round of NSS.

STATA (version 13.0) statistical analysis software [34], after adjusting the survey design and the sampling weight with *svyset* command, for three rounds separately was used to carry out all the multivariable statistical analyses [35]. The survey design has remained uniform, and state and national level sample weights, named multipliers, have already been provided in the NSS data files for all three NSS rounds. Therefore, one can compare these results representative at state and national levels over the rounds. The algorithm for arriving the analytical sample for each round is given in Fig 1.

## Results

### Socioeconomic profile of the persons who reported any morbidity

The socioeconomic and demographic characteristics of individuals affected by non-communicable diseases (NCD) included in the three NSS rounds are shown in Table 1. Out of the total sample, 48% were males in 2004 compared to around 69% in 2018. The percentage of patients aged 60 years and above showed a steady increase from the first round (26%) to the third one (32%). Seventy-two percent of the sampled population lived in rural areas in 2004, while in 2018, only 52% of the population did so. In the first and the second rounds, about 50% and 30% population, respectively, had no education compared with 35% in the third round. The percentage of individuals who attained secondary and above education was higher in the third NSS round (27%) than in the earlier two rounds. In 2004 and 2018, the population in the poorest MPCE quintile was around 20%, which decreased to 13% in 2018. In 2004, the highest percentage of individuals was from the South region, whereas in 2018, the highest percentage was from the West region.

### Variation in unmet need for treatment-seeking behaviour

**State-wise variation.** There were seven states/union-territories, namely Andhra Pradesh, Daman and Diu, Gujarat, Haryana, Maharashtra, Punjab, and Uttar Pradesh, where the unmet

**Table 1. Profile of individuals who have suffered/are suffering from any disease in 15 days prior to the survey date by sociodemographic characteristics in India, NSS, 2004–2018.**

| Sociodemographic characteristics | NSS-2004 (n = 38,753) | | NSS-2014 (n = 37,282) | | NSS-2018 (n = 43,239) | |
|---|---|---|---|---|---|---|
| | n | % | n | % | n | % |
| **Age (in years)** | | | | | | |
| 0–14 | 4,786 | 14.4 | 7,132 | 19.5 | 7,992 | 18.5 |
| 15–35 | 3,975 | 12.6 | 6,838 | 18.5 | 6,534 | 15.1 |
| 36–59 | 17,328 | 47.0 | 12,941 | 36.0 | 15,006 | 34.7 |
| 60 and above | 12,664 | 26.0 | 10,258 | 26.0 | 13,684 | 31.7 |
| **Gender** | | | | | | |
| Male | 18,790 | 48.1 | 24,383 | 68.6 | 20,387 | 47.2 |
| Female | 19,963 | 51.9 | 12,071 | 31.4 | 22,830 | 52.8 |
| **Education** | | | | | | |
| Illiterate | 18,742 | 50.4 | 10,003 | 30.2 | 14,934 | 34.6 |
| Up to Primary | 9,965 | 26.0 | 9,464 | 26.2 | 11,348 | 26.3 |
| Middle | 8,494 | 20.4 | 10,242 | 27.6 | 5,224 | 12.1 |
| Secondary and above | 1,522 | 3.3 | 6,744 | 16.0 | 11,712 | 27.1 |
| **Marital Status** | | | | | | |
| Never married | 11,733 | 34.6 | 2,710 | 7.5 | 10,937 | 25.3 |
| Currently married | 20,424 | 50.4 | 30,194 | 82.5 | 25,696 | 59.5 |
| Others | 6,596 | 15.0 | 3,550 | 10.0 | 6,585 | 15.2 |
| **Relation to household's head** | | | | | | |
| Self | 12,564 | 30.9 | 24,053 | 70.1 | 15,388 | 35.6 |
| Spouse of head | 7,882 | 20.5 | 7,013 | 18.0 | 10,411 | 24.1 |
| Unmarried child | 8,197 | 25.6 | 1,864 | 5.4 | 7,048 | 16.3 |
| Married child | 1,057 | 2.2 | 1,817 | 3.2 | 1,162 | 2.7 |
| Spouse of child | 1,092 | 2.4 | 808 | 1.5 | 1,241 | 2.9 |
| Others | 7,961 | 18.5 | 899 | 1.8 | 7,968 | 18.4 |
| **Religion** | | | | | | |
| Hindu | 29,985 | 79.4 | 27,631 | 79.1 | 31,573 | 73.1 |
| Muslim | 5,387 | 13.4 | 5,510 | 13.4 | 7,321 | 16.9 |
| Others | 3,380 | 7.2 | 3,313 | 7.5 | 4,324 | 10.0 |
| **Caste** | | | | | | |
| SCs/STs | 9,071 | 24.1 | 8,801 | 23.8 | 9,757 | 22.6 |
| OBC | 14,883 | 38.8 | 15,336 | 44.6 | 17,400 | 40.3 |
| General | 14,792 | 37.1 | 12,317 | 31.6 | 16,061 | 37.2 |
| **MPCE quintile** | | | | | | |
| Poorest | 6,554 | 20.8 | 4,739 | 20.0 | 5,671 | 13.1 |
| Poorer | 5,531 | 17.1 | 6,548 | 21.7 | 6,470 | 15.0 |
| Middle | 6,686 | 18.7 | 6,615 | 19.5 | 7,590 | 17.6 |
| Richer | 8,434 | 20.4 | 8,867 | 20.9 | 10,151 | 23.5 |
| Richest | 11,548 | 22.9 | 9,684 | 17.9 | 13,336 | 30.9 |
| **Place of residence** | | | | | | |
| Rural | 23,971 | 71.6 | 18,825 | 62.8 | 22,373 | 51.7 |
| Urban | 14,782 | 28.4 | 18,457 | 37.2 | 20,866 | 48.3 |
| **Regions of residence** | | | | | | |
| North | 10,127 | 25.6 | 4,661 | 10.4 | 7,357 | 17.0 |
| West | 6,360 | 18.8 | 9,218 | 22.4 | 11,834 | 27.4 |
| East | 6,249 | 18.9 | 7,066 | 22.9 | 8,159 | 18.9 |

*(Continued)*

**Table 1.** (Continued)

| Sociodemographic characteristics | NSS-2004 (n = 38,753) | | NSS-2014 (n = 37,282) | | NSS-2018 (n = 43,239) | |
|---|---|---|---|---|---|---|
| | n | % | n | % | n | % |
| Northeast | 2,724 | 2.9 | 1,453 | 1.3 | 1,626 | 3.8 |
| South | 11,200 | 28.2 | 9,090 | 27.0 | 7,696 | 17.8 |
| Central | 2,093 | 5.8 | 5,794 | 16.0 | 6,567 | 15.2 |
| **India** | **38,753** | **100.0** | **37,282** | **100.0** | **43,239** | **100.0** |

Source: Authors' computation based on NSS data.

Note: All 'n' are unweighted. Total percentage may not be equal to 100 due to some missing cases, SCs/STs = Scheduled Castes/Scheduled Tribes, OBC = Other Backward Class, NSS = National Sample Survey, MPCE = Monthly Per Capita Expenditure.

need for treatment-seeking from public health facilities was more than 60%. Punjab, Tripura, and Uttar Pradesh were the three states that showed a continuous increase in the unmet need for treatment-seeking from public health facilities across three rounds. In 2004, the unmet need for treatment-seeking from public health facilities was the highest in Daman and Diu, while in 2018, it was in Punjab. In 2004, Meghalaya, with the two-fifths population (40%), and Nagaland, with nil proportion of the population, had the highest and the lowest proportion respectively of unmet need for treatment-seeking among all states/UTs. In 2018, Nagaland (27%) had the highest and Lakshadweep (0.0%) the lowest percentage of unmet need for treatment-seeking among states/UTs (S3 Table).

**Sociodemographic variation.** The unmet need for treatment-seeking decreased by 9.1 percentage points between 2004 and 2018. In 2018, the unmet need for treatment-seeking was the lowest in the age group 60 years and above compared with the other age groups. The three NSS rounds showed that the unmet need for treatment-seeking decreased with increasing education and MPCE. People in the urban areas had a lower unmet need for treatment-seeking than their rural counterparts (Table 2).

On the other side, the unmet need for treatment-seeking from public health facilities showed a large gap between the age groups 0–14 and 60 and above in 2004, which has almost disappeared in 2018. The unmet need for treatment-seeking from public health facilities in the age group 0–14 had a decreasing trend, while for the age group 15–35 years, there was an increasing trend from 2004 onwards. For all three rounds, the unmet need for treatment-seeking from public health facilities among people having secondary and above education was higher than that among non-literate respondents. In all three rounds of the survey, respondents from the other castes had a higher unmet need for treatment-seeking from the public health system than scheduled castes/scheduled tribes (SCs/STs) and other backward class (OBC). The unmet need for treatment-seeking from public health facilities increased with the MPCE quintile, evidently found in all three rounds. Urban residents had a higher unmet need than their rural counterparts. In 2004, the unmet need for treatment-seeking from public health facilities was the highest in the West region, while in 2018, this was so in the Central region (Table 2).

**Region-wise variation.** In 2004 and 2018, the unmet need for treatment-seeking was the highest in the Eastern region, whereas in 2014, it was the highest in the Northern region (Figs 2 and 3).

**Morbidity-wise variation.** The unmet need for treatment-seeking from public health facilities was the highest for endocrine disorders (77%) in 2004 and blood disorders (68%) in 2018. The unmet need for treatment-seeking from public health facilities to treat ear-related problems increased from 24% to 55% between 2004 and 2018. The unmet need for treatment-seeking decreased across the disease groups from 12% in 2004 to 3% in 2018 (Table 3).

**Table 2. Unmet need for treatment-seeking and unmet need for treatment-seeking from public health facilities among those who suffered/are suffering from any disease by Sociodemographic characteristics in India, NSS, 2004–2018.**

| Sociodemographic characteristics | Unmet need for treatment-seeking from any source (%) | | | Unmet need for treatment-seeking from public health facilities (%) | | |
|---|---|---|---|---|---|---|
| | NSS-2004 | NSS-2014 | NSS-2018 | NSS 2004 | NSS-2014 | NSS-2018 |
| **Age (in years)** | $\chi2 = 140.34^*$ | $\chi2 = 315.70^*$ | $\chi2 = 29.59^*$ | $\chi2 = 175.57^*$ | $\chi2 = 34.58^*$ | $\chi2 = 11.82^*$ |
| 0–14 | 9.0 | 15.4 | 3.4 | 67.7 | 63.6 | 62.7 |
| 15–35 | 13.5 | 14.9 | 2.5 | 57.5 | 60.4 | 61.3 |
| 36–59 | 10.7 | 9.8 | 2.7 | 60.9 | 63.1 | 61.8 |
| 60 and above | 14.3 | 7.7 | 2.1 | 53.6 | 64.9 | 61.0 |
| **Gender** | $\chi2 = 3.74\#$ | $\chi2 = 1.120ns$ | $\chi2 = 11.81^*$ | $\chi2 = 2.95ns$ | $\chi2 = 7.28^*$ | $\chi2 = 4.23\#$ |
| Male | 11.5 | 11.5 | 3.1 | 60.1 | 63.5 | 61.8 |
| Female | 11.9 | 10.7 | 2.2 | 59.1 | 62.7 | 61.6 |
| **Education** | $\chi2 = 251.97^*$ | $\chi2 = 81.31^*$ | $\chi2 = 31.44^*$ | $\chi2 = 353.28^*$ | $\chi2 = 552.97^*$ | $\chi2 = 559.08^*$ |
| Illiterate | 14.0 | 13.0 | 3.0 | 56.0 | 56.6 | 58.3 |
| Up to Primary | 10.9 | 12.9 | 2.9 | 59.6 | 59.7 | 57.5 |
| Middle | 8.2 | 9.3 | 3.3 | 66.5 | 67.4 | 61.8 |
| Secondary and above | 4.8 | 8.4 | 1.6 | 71.9 | 74.4 | 71.3 |
| **Marital Status** | $\chi2 = 106.25^*$ | $\chi2 = 15.86^*$ | $\chi2 = 17.39^*$ | $\chi2 = 95.96^*$ | $\chi2 = 13.95^*$ | $\chi2 = 66.07^*$ |
| Never married | 11.4 | 15.1 | 3.2 | 61.7 | 59.6 | 62.2 |
| Currently married | 10.6 | 11.0 | 2.5 | 60.8 | 64.1 | 62.6 |
| Others | 16.2 | 10.2 | 2.1 | 50.7 | 58.7 | 57.5 |
| **Relation to household's head** | $\chi2 = 58.67^*$ | $\chi2 = 18.90^*$ | $\chi2 = 18.26^*$ | $\chi2 = 58.52^*$ | $\chi2 = 57.01^*$ | $\chi2 = 27.73^*$ |
| Self | 11.9 | 11.6 | 2.7 | 57.9 | 62.2 | 59.9 |
| Spouse of head | 10.4 | 10.2 | 2.2 | 61.4 | 64.9 | 63.4 |
| Unmarried child | 12.4 | 15.5 | 3.1 | 60.7 | 61.3 | 63.2 |
| Married child | 8.7 | 7.8 | 2.1 | 67.2 | 71.3 | 68.4 |
| Spouse of child | 8.4 | 5.5 | 2.9 | 63.7 | 72.7 | 63.1 |
| Others | 12.8 | 6.7 | 2.7 | 57.5 | 67.5 | 59.8 |
| **Religion** | $\chi2 = 7.62\#$ | $\chi2 = 0.95ns$ | $\chi2 = 13.32^*$ | $\chi2 = 59.00^*$ | $\chi2 = 28.44^*$ | $\chi2 = 53.52^*$ |
| Hindu | 11.9 | 11.4 | 2.8 | 59.3 | 62.7 | 61.2 |
| Muslim | 12.1 | 12.1 | 2.1 | 60.7 | 64.0 | 62.0 |
| Others | 8.7 | 8.4 | 1.7 | 61.1 | 67.0 | 65.6 |
| **Caste** | $\chi2 = 149.37^*$ | $\chi2 = 75.37^*$ | $\chi2 = 49.95^*$ | $\chi2 = 731.16^*$ | $\chi2 = 705.47^*$ | $\chi2 = 722.23^*$ |
| SCs/STs | 15.4 | 13.3 | 2.9 | 51.0 | 52.8 | 54.4 |
| OBC | 11.4 | 10.9 | 2.5 | 61.2 | 63.7 | 59.8 |
| General | 9.6 | 10.2 | 2.6 | 63.4 | 70.5 | 69.2 |
| **MPCE quintile** | $\chi2 = 649.66^*$ | $\chi2 = 350.78^*$ | $\chi2 = 207.92^*$ | $\chi2 = 985.26^*$ | $\chi2 = 1200.00^*$ | $\chi2 = 923.59^*$ |
| Poorest | 17.8 | 18.5 | 4.6 | 47.1 | 47.2 | 52.5 |
| Poorer | 14.4 | 12.3 | 3.7 | 54.8 | 60.4 | 59.2 |
| Middle | 11.5 | 11.5 | 2.1 | 61.1 | 62.9 | 58.9 |
| Richer | 9.7 | 6.9 | 2.1 | 63.3 | 70.9 | 61.8 |
| Richest | 6.3 | 6.6 | 1.4 | 69.9 | 76.0 | 71.8 |
| **Place of residence** | $\chi2 = 170.12^*$ | $\chi2 = 132.43^*$ | $\chi2 = 72.94^*$ | $\chi2 = 311.12^*$ | $\chi2 = 367.81^{***}$ | $\chi2 = 527.54^*$ |
| Rural | 13.2 | 13.3 | 3.4 | 56.6 | 59.0 | 58.2 |
| Urban | 7.9 | 7.9 | 1.3 | 67.2 | 70.3 | 67.8 |

(*Continued*)

**Table 2.** (Continued)

| Sociodemographic characteristics | Unmet need for treatment-seeking from any source (%) | | | Unmet need for treatment-seeking from public health facilities (%) | | |
|---|---|---|---|---|---|---|
| | NSS-2004 | NSS-2014 | NSS-2018 | NSS 2004 | NSS-2014 | NSS-2018 |
| **India** | **11.7** | **11.3** | **2.6** | **59.6** | **63.2** | **61.7** |

Source: Authors' computation based on NSS data, χ2 = chi-square, Significant level

*p<0.01

#p<0.05, ns = not significant.

SCs/STs = Scheduled Castes/Scheduled Tribes, OBC = Other Backward Class, NSS = National Sample Survey, MPCE = Monthly Per Capita Expenditure.

The state-wise variation in the unmet need for treatment-seeking from any public health facilities and unmet need for treatment-seeking from any sources by major groups of diseases like communicable, non-communicable and other diseases are also quite visible in S4 and S5 Tables.

## Determinants of unmet need for treatment-seeking from public health facilities

The results from the binary logistic regression show that females were 8% more likely to have an unmet need for treatment-seeking from public health facilities than males in 2004, while the sex differentials were insignificant in 2018. Compared with illiterate respondents, those with a primary level of education were 20% more likely to have an unmet need for treatment-seeking from public health facilities; the level reduced to 7% in 2018. Results for 2004 reveal no significant difference in unmet needs between Hindus and Muslims. In contrast, in 2018, Muslims were 15% less likely to have an unmet need for treatment-seeking from any public health facilities than Hindus. General caste persons were significantly more likely to have an unmet need for treatment-seeking from public health facilities compared with SCs/STs across all three rounds. It was also found that the unmet need for treatment-seeking from public health

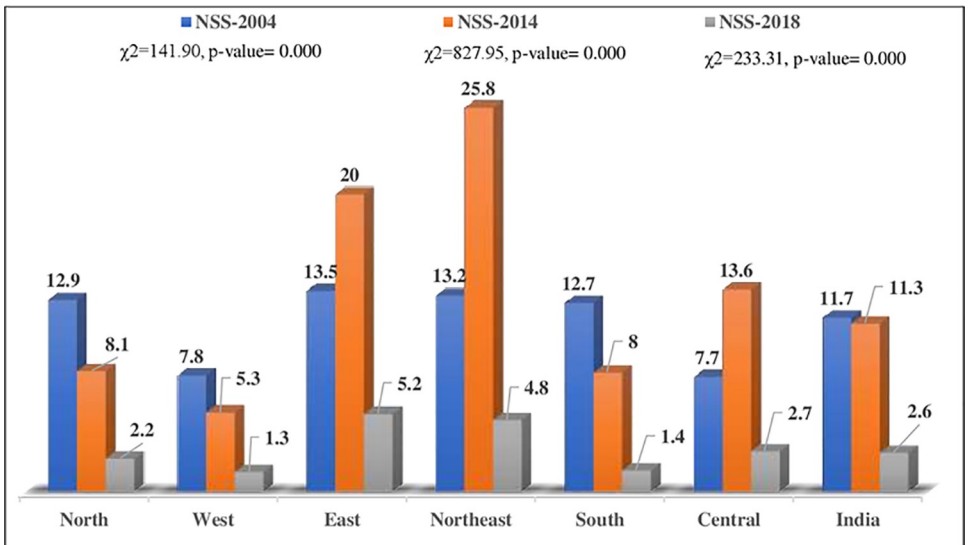

**Fig 2. Unmet need for treatment-seeking among those who suffered/are suffering from any disease by region of residence in India, NSS, 2004–2018.**

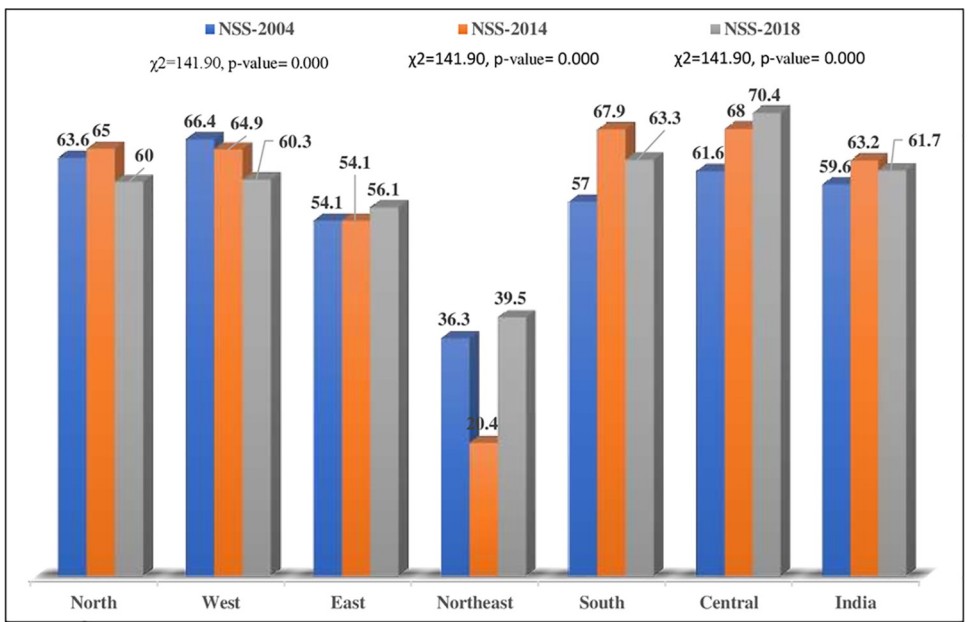

**Fig 3. Unmet need for treatment-seeking from public health facilities among those who suffered/are suffering from any disease by region of residence in India, NSS, 2004–2018.**

**Table 3. Unmet need for treatment-seeking from any source and treatment-seeking from public health facilities among who have suffered/are suffering from selected morbidities in India, NSS, 2004–2018.**

| Morbidities | Unmet need for treatment-seeking (%) from any source | | | Unmet need for treatment-seeking from public health facilities (%) | | |
|---|---|---|---|---|---|---|
| | **NSS-2004** | **NSS-2014** | **NSS-2018** | **NSS 2004** | **NSS-2014** | **NSS-2018** |
| Infections | 11.3 | 15.2 | 3.1 | 63.2 | 62.1 | 60.8 |
| Cancers | 3.9 | 3.2 | 0.4 | 55.2 | 57.9 | 49.6 |
| CVDs | 2.6 | 3.1 | 1.0 | 72.5 | 72.3 | 65.6 |
| Respiratory | 12.9 | 18.1 | 1.9 | 59.4 | 55.0 | 56.6 |
| Gastro–intestinal | 10.8 | 12.5 | 2.1 | 64.0 | 64.7 | 66.5 |
| Blood Disorders | 5.9 | 3.4 | 2.0 | 56.9 | 63.0 | 68.2 |
| Endocrine | 2.2 | 2.3 | 1.3 | 76.5 | 73.6 | 64.8 |
| Psychiatric | 11.5 | 16.8 | 6.5 | 51.8 | 53.9 | 53.5 |
| Injuries | 8.6 | 11.0 | 5.2 | 60.6 | 60.2 | 63.5 |
| Eye | 24.8 | 10.7 | 2.7 | 36.0 | 58.6 | 50.7 |
| Ear | 38.1 | 12.5 | 1.3 | 23.9 | 60.6 | 55.0 |
| Skin | 13.3 | 9.1 | 8.5 | 56.7 | 65.9 | 67.6 |
| Musculo-skeletal | 17.0 | 12.4 | 3.9 | 53.3 | 56.7 | 57.9 |
| Genito–urinary | 7.7 | 6.5 | 2.9 | 64.9 | 74.4 | 65.9 |
| Obstetrics | - | 6.3 | 0.3 | - | 67.9 | 61.0 |
| Others | 14.5 | 5.8 | 3.7 | 54.8 | 64.0 | 67.3 |
| **India** | **11.7** | **11.3** | **2.6** | **59.6** | **63.2** | **61.7** |

Source: Authors' computation based on NSS data. NSS = National Sample Survey.

facilities decreased significantly with a rise in the ladder of the MPCE quintile. In 2018, urban residents were 17% more likely to have the unmet need for treatment-seeking from public health facilities compared with their rural counterparts. Such an unmet need were two times more likely to be found in the Central region than in the Northern region (Table 4).

### Reasons for no treatment in three rounds of NSS

The main reason for not seeking any treatment was gathered in all three surveys. No treatment taken due to no health facility available in the neighbourhood has increased from 12.7% in 2004 to 15.4% in 2014 and declined to 8.7% in 2018. No treatment due to the financial problem has consistently decreased between 2004 and 2018. On the contrary, no treatment due to long waiting has increased from 1.0% in 2004 to 5.7% in 2014. In all three rounds, the ailment not considered serious enough was the main reason for not seeking treatment. It is to be noted that the proportion of respondents who did not seek treatment from any source, thinking ailment not so serious, has been showing an increasing trend from 36% to 71% between 2004 and 2018 (Table 5).

### Variation in reasons for unmet need for treatment-seeking from public health facilities

**Sociodemographic variations.** The unmet need for treatment-seeking from public health facilities due to financial constraints was higher among females than males in 2014 and 2018. The proportion of respondents reporting quality as the main reason for not seeking treatment from public health facilities has decreased from 47% in 2004 to 27% in 2018. In 2014 and 2018, the percentage of illiterate people reporting quality as the main reason for the unmet need for treatment-seeking from the public health facilities was higher than the rest of the educational subgroups. In all three rounds, the percentage of currently married respondents reporting quality as the main reason for the unmet need for treatment-seeking was much higher than those in other types of marital union. In 2018, compared with the general caste, a higher percentage of SCs/STs reported financial constraints as the main reason for the unmet need for treatment-seeking even from public health facilities (Table 6).

### Reasons for unmet need for treatment-seeking from public facilities by selected morbidities

Quality was cited as the main reason for the unmet need for treatment-seeking from public health facilities, the most by those suffering from genito-urinary diseases (55.6%) in 2004 and 41.0% in 2018. The lowest proportion of respondents suffering from ear-related ailments (36.2% in 2004 and 20.2% in 2018) followed by those suffering from eye-related ailments (49.3% in 2004 and 18.2% in 2018) said the quality of services at public health facilities was the main reason. Long waiting time was also cited as the main reason for unmet need for treatment-seeking from public health facilities the most by those suffering from blood disorders, 16.3% in 2004 and 27.2% 2018 and the least by those suffering from psychiatric diseases (14.9%) in 2004 and obstetric diseases (5.3%) in 2018 (Table 7).

### Determinants of main reason for unmet need for treatment-seeking from public health facilities

The relative risk ratio from the younger (0.51) to the older (0.47) age groups decreased more significantly for waiting time as the main reason for not availing treatment from public health

**Table 4. Adjusted odds ratios obtained from logistic regression analysis to examine effects of Sociodemographic characteristics on unmet need for treatment-seeking from public health facilities, India, NSS, 2004–2018.**

| Sociodemographic Characteristics | NSS-2004 | | NSS-2014 | | NSS-2018 | |
|---|---|---|---|---|---|---|
| | AOR | 95% CI | AOR | 95% CI | AOR | 95% CI |
| **Age (in years)** | | | | | | |
| 0–14 ref. | | | | | | |
| 15–35 | 0.58* | [0.52–0.64] | 0.82* | [0.76–0.88] | 0.59* | [0.54–0.66] |
| 36–59 | 0.47* | [0.42–0.52] | 0.81* | [0.76–0.87] | 0.61* | [0.54–0.69] |
| 60 and above | 0.42* | [0.38–0.47] | 0.83* | [0.78–0.89] | 0.62* | [0.54–0.7] |
| **Gender** | | | | | | |
| Male ref. | | | | | | |
| Female | 1.08* | [1.02–1.15] | 1.05ns | [0.96–1.16] | 0.96ns | [0.9–1.02] |
| **Education** | | | | | | |
| Illiterate ref. | | | | | | |
| Up to Primary | 1.20* | [1.14–1.27] | 1.01ns | [0.95–1.07] | 0.93* | [0.88–0.98] |
| Middle | 1.40* | [1.31–1.49] | 1.16* | [1.09–1.24] | 1.06ns | [0.99–1.14] |
| Secondary and above | 1.75* | [1.54–1.99] | 1.49* | [1.38–1.61] | 1.42* | [1.33–1.52] |
| **Marital Status** | | | | | | |
| Never married ref. | | | | | | |
| Currently married | 1.12*ns* | [0.99–1.27] | 1.29* | [1.05–1.58] | 1.17# | [1.02–1.34] |
| Others | 1.10*ns* | [0.96–1.25] | 1.25# | [1.01–1.55] | 1.04ns | [0.9–1.2] |
| **Relation to household's head** | | | | | | |
| Self ref. | | | | | | |
| Spouse of head | 1.07ns | [0.98–1.16] | 0.94ns | [0.85–1.05] | 1.15* | [1.06–1.25] |
| Unmarried child | 0.90*ns* | [0.80–0.10] | 1.02ns | [0.82–1.28] | 1.04ns | [0.93–1.16] |
| Married child | 0.90*ns* | [0.79–1.03] | 0.86* | [0.77–0.96] | 1.13* | [0.98–1.29] |
| Spouse of child | 1.02*ns* | [0.88–1.18] | 0.91 | [0.76–1.09] | 1.34* | [1.16–1.55] |
| Others | 0.88* | [0.82–0.96] | 1.17ns | [0.96–1.44] | 1.09* | [1.01–1.17] |
| **Religion** | | | | | | |
| Hindu ref. | | | | | | |
| Muslim | 0.99*ns* | [0.93–1.06] | 0.89* | [0.84–0.95] | 0.85* | [0.8–0.89] |
| Others | 0.86* | [0.79–0.93] | 1.02ns | [0.94–1.11] | 1.08* | [1–1.16] |
| **Caste** | | | | | | |
| SCs/STs ref. | | | | | | |
| OBC | 1.50* | [1.42–1.59] | 1.37* | [1.29–1.45] | 1.27* | [1.2–1.34] |
| General | 1.52* | [1.43–1.61] | 1.64* | [1.54–1.74] | 1.65* | [1.55–1.74] |
| **MPCE quintile** | | | | | | |
| Poorest ref. | | | | | | |
| Poorer | 1.24* | [1.15–1.33] | 1.38* | [1.27–1.49] | 1.19* | [1.11–1.28] |
| Middle | 1.43* | [1.33–1.54] | 1.55* | [1.43–1.68] | 1.32* | [1.22–1.42] |
| Richer | 1.55* | [1.45–1.67] | 1.89* | [1.75–2.04] | 1.44* | [1.34–1.55] |
| Richest | 1.96* | [1.83–2.11] | 2.58* | [2.37–2.8] | 2.04* | [1.89–2.21] |
| **Place of residence** | | | | | | |
| Rural ref. | | | | | | |
| Urban | 1.05** | [1.01–1.11] | 1.12* | [1.07–1.17] | 1.17* | [1.12–1.23] |
| **Regions of residence** | | | | | | |
| North ref. | | | | | | |
| West | 1.14* | [1.07–1.22] | 1.25* | [1.16–1.35] | 1.21* | [1.14–1.29] |
| East | 0.81* | [0.75–0.86] | 1.00ns | [0.92–1.08] | 1.32* | [1.23–1.41] |

*(Continued)*

**Table 4.** (Continued)

| Sociodemographic Characteristics | NSS-2004 | | NSS-2014 | | NSS-2018 | |
|---|---|---|---|---|---|---|
| | AOR | 95% CI | AOR | 95% CI | AOR | 95% CI |
| Northeast | 0.33* | [0.30–0.36] | 0.30* | [0.26–0.34] | 0.41* | [0.36–0.46] |
| South | 0.92* | [0.87–0.98] | 1.50* | [1.39–1.63] | 1.54* | [1.44–1.65] |
| Central | 1.04ns | [0.94–1.14] | 1.73* | [1.59–1.89] | 2.29* | [2.13–2.48] |
| **Constants** | 1.14# | [1.01–1.30] | 0.50* | [0.40–0.63] | 0.71* | [0.63–0.81] |

Source: Authors' computation based on NSS data; ref = Reference category; **AOR** = Adjusted Odds Ratio; Significant level

*p<0.01

#p<0.05, ns = not significant.

MPCE = Monthly Per Capita Expenditure, SCs/STs = Scheduled Castes/ Scheduled Tribes, OBC = Other Backward Class, NSS = National Sample Survey.

facilities compared with the unavailability of the required services. The relative risk ratio declined when the educational level of respondents moved up from the illiterate to middle (RR = 0.76) for quality as the main reason for unmet need compared to unavailability of required services. When the MPCE quintile changed from the poorest to the middle, the relative risk ratio increased for quality as the main reason for not seeking treatment from public health facilities compared with the unavailability of the required services. The relative risk ratio for quality as the main reason for the unmet need for treatment-seeking increased from 1.37 for rural residents to 1.52 for urban residents. On the other hand, the relative risk ratio for financial constraints as the prime reason for no treatment-seeking from public health facilities was more than three times higher in the East and Northeast region than in the North region. The estimated relative risk ratio for financial constraints as the main reason against the unavailability of required services for the unmet need in treatment-seeking from public facilities was significantly higher among Muslims (2.55 times) than Hindus. The relative risk ratio of reporting long waiting time as the main reason for not seeking treatment from public health facilities increased four times while moving from the poorest to the middle MPCE quintile (Table 8).

## Discussion

This study examined the sociodemographic, geographic and selected morbidities variations in the unmet need for treatment-seeking in public health facilities and the reasons thereof. This study also analyzed the sociodemographic determinants of seeking no treatment from any kind of health facility, either public or private. The uniqueness of this study is that the study's findings are generalizable as it used three rounds of a nationally-representative sample survey covering all the states and union territories of India. This study dealt with aspects that are rarely available from the clients' perspectives across states/union territories, sociodemographic factors, and selected diseases.

**Table 5. Reasons (percentage distribution) for no treatment-seeking in three rounds of NSS, India, 2004–2018.**

| Main reason for no treatment-seeking | 2004 | 2014 | 2018 |
|---|---|---|---|
| | Percent (N = 4640) | Percent (N = 3431) | Percent (N = 3174) |
| No facility available in neighbourhood | 12.7 | 15.4 | 8.7 |
| Financial problem | 31.2 | 6.1 | 2.7 |
| Long waiting | 1.0 | 3.4 | 5.7 |
| Ailment not considered serious | 35.5 | 57.4 | 70.7 |
| Others | 19.7 | 17.7 | 12.2 |

**Table 6. Reasons for unmet need for treatment-seeking from public health facilities by sociodemographic characteristics, India, NSS, 2004–2018.**

| Sociodemographic Characteristics | Quality | | | Waiting | | | Financial | | | Others | | |
|---|---|---|---|---|---|---|---|---|---|---|---|---|
| | NSS-2004[#] | NSS-2014 | NSS-2018 | NSS-2004[#] | NSS-2014 | NSS-2018 | NSS-2004[#] | NSS-2014 | NSS-2018 | NSS-2004[#] | NSS-2014 | NSS-2018 |
| **Age (in years)** | | | | | | | | | | | | |
| 0–14 | 46.5 | 39.5 | 27.3 | 10.9 | 28.0 | 15.3 | NA | 0.5 | 0.3 | 42.6 | 32.0 | 57.1 |
| 15–35 | 42.1 | 45.3 | 27.9 | 12.0 | 23.3 | 16.3 | NA | 0.3 | 0.5 | 46.0 | 31.1 | 55.3 |
| 36–59 | 47.2 | 41.4 | 28.4 | 12.4 | 27.4 | 17.6 | NA | 0.5 | 0.8 | 40.5 | 30.8 | 53.3 |
| 60 and above | 48.6 | 45.1 | 25.4 | 11.7 | 29.6 | 19.4 | NA | 0.6 | 0.3 | 39.8 | 24.7 | 54.9 |
| **Gender** | | | | | | | | | | | | |
| Male | 47.4 | 44.3 | 28.1 | 11.5 | 26.1 | 17.2 | NA | 0.4 | 0.3 | 41.2 | 29.2 | 54.4 |
| Female | 46.2 | 39.8 | 26.5 | 12.4 | 29.1 | 17.6 | NA | 0.6 | 0.7 | 41.5 | 30.6 | 55.2 |
| **Education** | | | | | | | | | | | | |
| Illiterate | 47.6 | 45.4 | 29.1 | 9.9 | 21.3 | 16.4 | NA | 0.8 | 0.8 | 42.6 | 32.6 | 53.7 |
| Up to Primary | 44.1 | 41.0 | 26.0 | 12.7 | 29.4 | 17.9 | NA | 0.7 | 0.5 | 43.3 | 28.9 | 55.6 |
| Middle | 47.6 | 41.6 | 26.8 | 14.7 | 27.7 | 16.6 | NA | 0.2 | 0.1 | 37.7 | 30.5 | 56.5 |
| Secondary and above | 50.4 | 43.8 | 26.4 | 15.3 | 31.2 | 18.5 | NA | 0.2 | 0.4 | 34.3 | 24.8 | 54.7 |
| **Marital Status** | | | | | | | | | | | | |
| Never married | 43.9 | 43.8 | 26.5 | 11.7 | 31.3 | 15.2 | NA | 0.1 | 0.4 | 44.4 | 24.9 | 57.9 |
| Currently married | 49.0 | 42.8 | 28.2 | 11.7 | 26.9 | 18.3 | NA | 0.5 | 0.4 | 39.3 | 29.9 | 53.2 |
| Others | 45.6 | 42.9 | 25.3 | 13.4 | 25.2 | 18.3 | NA | 1.2 | 1.2 | 41.0 | 30.8 | 55.3 |
| **Relation to household's head** | | | | | | | | | | | | |
| Self | 47.0 | 43.7 | 27.3 | 12.5 | 25.6 | 18.6 | NA | 0.6 | 0.7 | 40.5 | 30.1 | 53.5 |
| Spouse of head | 48.4 | 39.6 | 28.2 | 12.2 | 29.9 | 18.1 | NA | 0.5 | 0.3 | 39.4 | 30.1 | 53.4 |
| Unmarried child | 42.9 | 44.6 | 25.6 | 11.6 | 31.1 | 14.0 | NA | 0.1 | 0.4 | 45.6 | 24.3 | 60.1 |
| Married child | 51.4 | 45.9 | 30.4 | 11.6 | 29.3 | 12.2 | NA | 0.3 | 0.2 | 37.1 | 24.5 | 57.2 |
| Spouse of child | 51.2 | 43.1 | 30.4 | 10.1 | 31.2 | 18.7 | NA | 0.1 | 0.3 | 38.7 | 25.6 | 50.6 |
| Others | 48.9 | 35.4 | 27.1 | 11.4 | 29.7 | 18.9 | NA | 0.1 | 0.7 | 39.7 | 34.8 | 53.4 |
| **Religion** | | | | | | | | | | | | |
| Hindu | 47.6 | 44.6 | 28.2 | 11.9 | 25.3 | 17.3 | NA | 0.6 | 0.6 | 40.5 | 29.6 | 53.9 |
| Muslim | 42.8 | 37.1 | 26.3 | 12.0 | 30.6 | 17.0 | NA | 0.3 | 0.3 | 45.2 | 32.1 | 56.4 |
| Others | 45.3 | 36.6 | 19.3 | 11.7 | 37.5 | 19.5 | NA | 0.1 | 0.1 | 43.0 | 25.8 | 61.1 |
| **Caste** | | | | | | | | | | | | |
| SCs/STs | 43.4 | 38.8 | 25.3 | 11.2 | 25.3 | 18.0 | NA | 0.6 | 1.3 | 45.4 | 35.4 | 55.4 |
| OBC | 48.6 | 43.6 | 29.6 | 11.1 | 28.8 | 15.4 | NA | 0.3 | 0.2 | 40.2 | 27.3 | 54.8 |
| General | 46.6 | 44.4 | 25.8 | 13.1 | 25.7 | 19.3 | NA | 0.6 | 0.4 | 40.3 | 29.3 | 54.6 |
| **MPCE quintile** | | | | | | | | | | | | |
| Poorest | 39.2 | 42.3 | 23.1 | 9.2 | 22.0 | 12.0 | NA | 1.0 | 1.3 | 51.6 | 34.7 | 63.6 |
| Poorer | 41.2 | 45.3 | 28.2 | 10.3 | 22.9 | 16.6 | NA | 1.0 | 0.8 | 48.4 | 30.9 | 54.5 |
| Middle | 48.4 | 42.5 | 32.3 | 10.8 | 28.1 | 15.8 | NA | 0.3 | 0.4 | 40.8 | 29.2 | 51.5 |
| Richer | 49.2 | 44.3 | 28.8 | 13.4 | 27.4 | 19.4 | NA | 0.2 | 0.2 | 37.4 | 28.1 | 51.6 |
| Richest | 51.5 | 39.9 | 24.8 | 14.1 | 33.1 | 20.2 | NA | 0.1 | 0.3 | 34.5 | 26.9 | 54.8 |
| **Place of residence** | | | | | | | | | | | | |
| Rural | 46.1 | 43.7 | 28.6 | 9.4 | 22.8 | 14.9 | NA | 0.4 | 0.7 | 44.5 | 33.1 | 55.9 |
| Urban | 48.1 | 41.4 | 25.3 | 17.3 | 33.8 | 21.2 | NA | 0.6 | 0.3 | 34.6 | 24.3 | 53.3 |
| **Regions of residence** | | | | | | | | | | | | |
| North | 51.2 | 38.3 | 26.9 | 10.5 | 33.8 | 23.7 | NA | 0.8 | 0.1 | 38.3 | 27.1 | 49.3 |
| West | 52.9 | 34.6 | 20.2 | 10.5 | 35.3 | 17.2 | NA | 0.4 | 0.3 | 36.6 | 29.7 | 62.3 |
| East | 37.8 | 38.4 | 21.9 | 15.1 | 24.5 | 18.8 | NA | 0.9 | 1.1 | 47.2 | 36.3 | 58.3 |

*(Continued)*

**Table 6.** (Continued)

| Sociodemographic Characteristics | Quality | | | Waiting | | | Financial | | | Others | | |
|---|---|---|---|---|---|---|---|---|---|---|---|---|
| | NSS-2004# | NSS-2014 | NSS-2018 | NSS-2004# | NSS-2014 | NSS-2018 | NSS-2004# | NSS-2014 | NSS-2018 | NSS-2004# | NSS-2014 | NSS-2018 |
| Northeast | 23.2 | 27.8 | 18.9 | 7.4 | 25.2 | 16.9 | NA | 0.5 | 0.4 | 69.4 | 46.5 | 63.9 |
| South | 45.8 | 50.3 | 34.7 | 14.4 | 26.2 | 21.0 | NA | 0.3 | 0.2 | 39.8 | 23.3 | 44.2 |
| Central | 42.0 | 48.9 | 33.8 | 4.2 | 17.9 | 10.3 | NA | 0.3 | 0.7 | 53.9 | 32.9 | 55.2 |
| **India** | **46.8** | **42.7** | **27.3** | **11.9** | **27.4** | **17.4** | NA | **0.5** | **0.5** | **41.3** | **29.5** | **54.8** |

Source: Authors' computation based on NSS data

# Financial reasons were not collected in the NSS 2004 round.

MPCE = Monthly Per Capita Expenditure, SCs/STs = Scheduled Castes/Scheduled Tribes, OBC = Other Backward Class, NSS = National Sample Survey.

One of the critical findings of this study is that the main reason for not seeking treatment from any sources was the ailment not being considered serious enough, which supports the results of the previous studies [36,37]. Ghosh [37] found that the proportion of patients who did not seek medical treatment due to non-availability of medical facilities went up from 12.7% in 2004 to 15.4% in 2014 in rural areas that later decreased to 8.7% in 2018. Our results also indicate and support evidence from the fourth round of National Family Health Survey (NFHS-4), 2015–16, which reveals the self-reported unmet need for treating selected diseases in the age group 15–49 was higher among men than women. In NFHS-4, the estimated level of unmet need for treatment-seeking among men and women was respectively 65% and 34% for any cancer, 39% and 29% for Asthma, 39% and 28% for any heart disease, 27% and 19% for

**Table 7.** Main reason for unmet need for treatment-seeking from public health facilities by type of morbidities in the ICD-10, India, NSS, 2004–2018.

| Sociodemographic Characteristics | Quality | | | Waiting | | | Financial | | | Others | | |
|---|---|---|---|---|---|---|---|---|---|---|---|---|
| | NSS-2004# | NSS-2014 | NSS-2018 | NSS-2004# | NSS-2014 | NSS-2018 | NSS-2004# | NSS-2014 | NSS-2018 | NSS-2004# | NSS-2014 | NSS-2018 |
| Infections | 44.3 | 39.5 | 24.3 | 11.1 | 26.3 | 15.4 | NA | 0.4 | 0.4 | 44.6 | 33.8 | 59.9 |
| Cancers | 53.1 | 56.0 | 32.9 | 13.2 | 10.3 | 19.5 | NA | 3.4 | 1.7 | 33.7 | 30.4 | 45.9 |
| CVDs | 49.8 | 43.9 | 27.6 | 14.9 | 32.9 | 18.2 | NA | 0.5 | 0.2 | 35.4 | 22.7 | 54.0 |
| Respiratory | 50.4 | 42.0 | 30.6 | 12.3 | 28.1 | 18.3 | NA | 0.3 | 0.0 | 37.3 | 29.6 | 51.1 |
| Gastro–intestinal | 47.5 | 44.6 | 28.7 | 14.6 | 22.0 | 17.8 | NA | 1.0 | 1.7 | 37.9 | 32.5 | 51.8 |
| Blood disorders | 39.3 | 63.0 | 25.3 | 16.3 | 11.1 | 27.2 | NA | 0.0 | 6.3 | 44.4 | 25.9 | 41.2 |
| Endocrine | 53.6 | 45.0 | 24.9 | 14.1 | 32.4 | 19.0 | NA | 0.5 | 0.4 | 32.3 | 22.2 | 55.7 |
| Psychiatric | 45.9 | 48.2 | 31.2 | 7.4 | 20.0 | 11.3 | NA | 0.2 | 0.1 | 46.7 | 31.7 | 57.4 |
| Injuries | 41.0 | 37.3 | 33.6 | 14.9 | 27.4 | 25.2 | NA | 0.2 | 0.1 | 44.1 | 35.1 | 41.2 |
| Eye | 49.3 | 41.0 | 18.2 | 9.5 | 23.8 | 18.9 | NA | 0.2 | 0.4 | 41.3 | 35.1 | 62.4 |
| Ear | 36.2 | 63.9 | 20.2 | 10.7 | 11.8 | 21.6 | NA | 0.0 | 0.0 | 53.1 | 24.2 | 58.3 |
| Skin | 50.3 | 42.7 | 38.8 | 8.8 | 25.8 | 13.6 | NA | 0.4 | 0.1 | 40.9 | 31.2 | 47.5 |
| Musculo-skeletal | 51.6 | 42.5 | 30.7 | 8.7 | 26.2 | 20.1 | NA | 1.0 | 1.3 | 39.7 | 30.2 | 47.9 |
| Genito–urinary | 55.6 | 43.9 | 41.0 | 10.4 | 17.6 | 17.3 | NA | 0.3 | 1.4 | 33.9 | 38.2 | 40.3 |
| Obstetrics | - | 36.9 | 46.3 | - | 35.1 | 5.3 | NA | 0.0 | 0.0 | - | 28.1 | 48.5 |
| Others | 43.6 | 36.6 | 23.5 | 12.3 | 28.2 | 21.3 | NA | 0.1 | 0.2 | 44.1 | 35.1 | 55.1 |
| **All** | **46.8** | **42.7** | **27.3** | **11.9** | **27.4** | **17.4** | NA | **0.5** | **0.5** | **41.3** | **29.5** | **54.8** |

Source: Authors' computation based on NSS data 2004–2018

# Financial reasons were not collected in the NSS 2004 round.

CVDs = cardiovascular diseases.

**Table 8. Results from the multinomial regression models to examine the effect of sociodemographic characteristics on Unmet need for treatment-seeking for public health facilities, India, NSS, 2018.**

| Sociodemographic Characteristics | Quality | | Waiting | | Financial | | Others | |
|---|---|---|---|---|---|---|---|---|
| | RRR | 95% CI | RRR | 95% CI | RRR | 95% CI | RRR | 95% CI |
| **Age (in years)** | | | | | | | | |
| 0–14 ref. | | | | | | | | |
| 15–35 | 0.94ns | [0.73–1.21] | 0.81 ns | [0.6–1.08] | 2.31# | [0.99–5.35] | 0.83 ns | [0.65–1.06] |
| 36–59 | 0.81 ns | [0.60–1.10] | 0.51* | [0.35–0.73] | 2.9# | [1.05–7.97] | 0.75# | [0.56–1] |
| 60 and above | 0.82 ns | [0.60–1.12] | 0.47* | [0.32–0.68] | 1.44 ns | [0.49–4.25] | 0.83 ns | [0.62–1.12] |
| **Gender** | | | | | | | | |
| Male ref. | | | | | | | | |
| Female | 0.86ns | [0.74–1.01] | 0.98 ns | [0.82–1.17] | 1.36 ns | [0.8–2.33] | 0.99 ns | [0.85–1.15] |
| **Education** | | | | | | | | |
| Illiterate ref. | | | | | | | | |
| Up to Primary | 0.80* | [0.69–0.91] | 0.85# | [0.73–1] | 0.76 ns | [0.46–1.25] | 0.93 ns | [0.82–1.06] |
| Middle | 0.76* | [0.63–0.90] | 0.73* | [0.59–0.91] | 0.69 ns | [0.35–1.35] | 0.89 ns | [0.75–1.05] |
| Secondary and above | 0.83# | [0.71–0.97] | 0.73* | [0.6–0.89] | 0.74 ns | [0.4–1.35] | 0.97 ns | [0.84–1.13] |
| **Marital Status** | | | | | | | | |
| Never married ref. | | | | | | | | |
| Currently married | 1.01 ns | [0.71–1.44] | 0.76 ns | [0.49–1.16] | 0.57 ns | [0.19–1.71] | 0.72 ns | [0.52–1.01] |
| Others | 0.99 ns | [0.68–1.43] | 1.03 ns | [0.66–1.61] | 0.50 ns | [0.16–1.61] | 0.83 ns | [0.58–1.17] |
| **Relation to household's head** | | | | | | | | |
| Self ref. | | | | | | | | |
| Spouse of head | 1.31* | [1.08–1.61] | 1.29# | [1.02–1.65] | 0.65 ns | [0.31–1.34] | 1.19 ns | [0.99–1.44] |
| Unmarried child | 1.14 ns | [0.86–1.53] | 0.91 ns | [0.65–1.29] | 0.80 ns | [0.31–2.09] | 0.80 ns | [0.61–1.06] |
| Married child | 1.05 ns | [0.76–1.44] | 0.78 ns | [0.51–1.18] | 0.59 ns | [0.13–2.64] | 1.21 ns | [0.9–1.63] |
| Spouse of child | 1.31 ns | [0.92–1.85] | 0.97 ns | [0.64–1.49] | 0.60 ns | [0.15–2.38] | 1.20 ns | [0.86–1.66] |
| Others | 1.28# | [1.05–1.56] | 0.87 ns | [0.67–1.12] | 1.87 ns | [0.96–3.62] | 1.04 ns | [0.86–1.25] |
| **Religion** | | | | | | | | |
| Hindu ref. | | | | | | | | |
| Muslim | 0.93 ns | [0.81–1.07] | 1.00 ns | [0.84–1.18] | 0.85 ns | [0.49–1.47] | 0.92 ns | [0.81–1.06] |
| Others | 0.76* | [0.64–0.91] | 0.87 ns | [0.7–1.09] | 1.19 ns | [0.6–2.38] | 0.95 ns | [0.81–1.12] |
| **Caste** | | | | | | | | |
| SCs/STs ref. | | | | | | | | |
| OBC | 1.04 ns | [0.90–1.20] | 0.92 ns | [0.78–1.08] | 0.68 ns | [0.42–1.09] | 1.10 ns | [0.96–1.26] |
| General | 1.11 ns | [0.95–1.28] | 0.88 ns | [0.74–1.05] | 0.64 ns | [0.38–1.06] | 1.04 ns | [0.91–1.2] |
| **MPCE quintile** | | | | | | | | |
| Poorest ref. | | | | | | | | |
| Poorer | 1.31* | [1.09–1.57] | 1.01 ns | [0.82–1.25] | 0.7 ns | [0.39–1.25] | 1.34* | [1.12–1.6] |
| Middle | 1.53* | [1.28–1.84] | 1.04 ns | [0.84–1.29] | 0.82 ns | [0.46–1.46] | 1.42* | [1.2–1.7] |
| Richer | 1.53* | [1.28–1.84] | 1.01 ns | [0.81–1.25] | 0.51# | [0.26–0.98] | 1.65* | [1.39–1.96] |
| Richest | 1.50* | [1.24–1.82] | 0.92 ns | [0.73–1.16] | 0.63 ns | [0.32–1.24] | 1.92* | [1.6–2.3] |
| **Place of residence** | | | | | | | | |
| Rural ref. | | | | | | | | |
| Urban | 1.37* | [1.22–1.54] | 0.87# | [0.75–1] | 1.24 ns | [0.81–1.92] | 1.52* | [1.36–1.69] |
| **Regions of residence** | | | | | | | | |
| North ref. | | | | | | | | |
| West | 0.59* | [0.50–0.70] | 0.54* | [0.44–0.66] | 1.29 ns | [0.59–2.82] | 1.13 ns | [0.97–1.31] |
| East | 0.93 ns | [0.77–1.12] | 1.09 ns | [0.88–1.35] | 3.34* | [1.57–7.11] | 1.41* | [1.18–1.68] |

*(Continued)*

**Table 8.** (Continued)

| Sociodemographic Characteristics | Quality | | Waiting | | Financial | | Others | |
|---|---|---|---|---|---|---|---|---|
| | RRR | 95% CI | RRR | 95% CI | RRR | 95% CI | RRR | 95% CI |
| Northeast | 0.54* | [0.39–0.76] | 0.27* | [0.17–0.44] | 3.28# | [1.19–9.04] | 0.73# | [0.54–0.99] |
| South | 1.00 ns | [0.84–1.19] | 0.46* | [0.37–0.58] | 1.01 ns | [0.43–2.37] | 0.82# | [0.7–0.97] |
| Central | 1.22# | [1.01–1.46] | 0.85 ns | [0.69–1.06] | 2.26# | [1.04–4.89] | 0.80* | [0.67–0.95] |
| **Constants** | 2.89* | [2.08–4.02] | 3.91* | [2.65–5.78] | 0.05* | [0.02–0.15] | 5.41* | [3.96–7.38] |

Source: Authors' computation based on NSS data; RRR = Relative Risk Ratio ref = Reference category; Significant level

*p<0.01

#p<0.05, ns = not significant.

MPCE = Monthly Per Capita Expenditure, SCs/STs = Scheduled Castes/Scheduled Tribes, OBC = Other Backward Class, NSS = National Sample Survey.

diabetes, 44% and 13% for goitre or any other thyroid disorder (International Institute for Population Sciences [15].

The present study provides evidence that India's distinct geographical pattern in public health facilities utilization prevails. It can be observed that compared with urban and plain regions, the use of public facilities was higher in hilly and scheduled areas (tribal areas), which could be due to the unavailability of private health facilities in these difficult terrains [38]. The finding by Kumar that three-fourths of health infrastructure is concentrated in urban areas only, where only 27% of Indian reside [38], supports our finding regarding the low usage of private health facilities in hilly and scheduled areas. Other studies also discovered that the geographical distribution of health care facilities has a significant effect on the utilization of services and needs to be taken into account to bring more equity in public health care utilization in India [39,40].

Among the social groups, other backward class (OBC) and general castes are more likely to get treatment from private sources of care than SCs/STs. A strong association between social status and the use of private sources of care has been reported on earlier occasions [41]. The health care utilization was found to be lower in most of the Economic Action Group (EAG) states and the Northeastern states. The proportion of patients who did not have access to medical care due to the unavailability of medical facilities was also higher in these regions. Though the average cost of treatment in private health facilities is much higher than in public facilities, the low utilization of public healthcare may be due to the perception that public facilities are not of good quality [38].

Our analysis in this study has clearly shown that more affluent and urban respondents were more concerned about the quality of services at the public health facilities than their respective counterparts. On the other hand, for respondents residing in the Western, Eastern, and Central regions, financial constraints were the main barrier to seeking treatment from public health facilities. Surprisingly, our findings from the 2018 dataset reveal that even the educated masses did not report either poor quality or long waiting time as a major barrier to treatment-seeking from public health facilities. This could be due to a common perception that acute shortage of human resources, lack of testing and diagnostic facilities, non-availability of medicines, and lack of other infrastructure at public health facilities, which is especially documented at the primary level [42]. However, quality of care at public health facilities was the most pressing concern among the economically better-off groups of respondents. In contrast, the respondents from poor and socially disadvantaged groups (SCs/STs) still report financial constraints in seeking treatment even from public health facilities. Such a reporting behaviour could be due to various reasons, including lack of awareness regarding free services at public health

facilities, high direct costs and indirect costs (other than drug and medical consulting fees) such as wage loss, transportation, companion's time, etc.

## Limitations of the study

Despite the fact that this study has several strengths, it does have a few data limitations. The diseases on which information was collected were not the same across the three survey rounds. Therefore, the prevalence of any morbidity across the three NSS rounds may not be strictly comparable. Also, the NSS collects data on reported morbidity and does not directly measure morbidity using any field/hospital-based clinical anthropometric biomarkers or diagnostic tools. On the day of the survey, the morbidity reported by respondents may be influenced by personal bias and awareness of respondents regarding signs and symptoms of about 160 diseases. These facts can sometimes affect the actual level of morbidity in the population. Moreover, the proportion of respondents reporting the main reason as 'Others' for unmet need for treatment-seeking from public health facilities has also grown significantly across diseases. Hence, this study could not find what could be the specific reason reported under the 'Others' category. In the future rounds, NSS should collect more categories of the main reason for not seeking treatment from public health facilities.

## Conclusions

This study concludes that sociodemographic, regional and disease-wise variations in unmet need for treatment-seeking from public health facilities or any health facility (public/private) are vast in India. It highlighted a few potential barriers in treatment-seeking. Our study also observed significant sociodemographic and regional variations in reporting quality and financial constraints as the main reason for not availing treatment from public health facilities. Hence, we suggest that improving the availability of various kinds of services at public health facilities should be the top priority in ensuring equitable access to affordable and assured quality health services for all Indian citizens. There must be an emphasis on mass media strategies for perception and behaviour change towards the possible severity of each ailment, knowing that early treatment-seeking can prevent infirmity, disability and mortality in future. In this regard, the provisions under the Pradhan Mantri Jan Arogya Yojna (PMJAY), popularly known as "Ayushman Bharat", a flagship scheme of the Government of India to achieve the vision of Universal Health Coverage (UHC) launched in 2017, must be closely monitored until the last person's health is protected. So, that India's commitment towards the Sustainable Development Goals' agenda of *leaving no one behind* can be achieved soon.

## Supporting information

**S1 Table. Disease classification and coding of the NSS data used in the analysis.** (DOCX)

**S2 Table. Description and Scale of measurements of explanatory variables.** (DOCX)

**S3 Table. State-wise unmet need for treatment seeking from any sources and unmet need for treatment seeking from any public health facilities among those who have suffered/suffering from any disease in India, NSS, 2004–2018.** (DOCX)

**S4 Table. States wise variation for unmet need for treatment seeking from any public health facilities among those who have not taken treatment from the any public health**

**facility by major diseases conditions in India, NSS 2004–2018, India.**
(DOCX)

**S5 Table. States wise variation for unmet need for treatment seeking from any sources among those who have not seeking treatment from any sources by major diseases conditions in India, NSS 2004–2018, India.**
(DOCX)

## Acknowledgments

This study used the National Sample Survey (2004, 2014, 2018) datasets. Hence, the authors gratefully acknowledge the Survey Coordination Division of MoSPI, Government of India, for making anonymized data available to the general public. The analysis and inferences drawn in this research article are solely those of the authors and do not necessarily reflect the official position of their respective affiliated institutions or governing bodies.

## Author Contributions

**Conceptualization:** Rajaram Yadav, Jeetendra Yadav, Chander Shekhar.

**Data curation:** Jeetendra Yadav.

**Formal analysis:** Rajaram Yadav.

**Methodology:** Jeetendra Yadav, Chander Shekhar.

**Supervision:** Jeetendra Yadav, Chander Shekhar.

**Validation:** Rajaram Yadav.

**Writing – original draft:** Jeetendra Yadav.

**Writing – review & editing:** Rajaram Yadav, Jeetendra Yadav, Chander Shekhar.

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
