## [Decision Letter · Decision Letter 0]

5 Oct 2021

PGPH-D-21-00530

Unmet need for treatment-seeking from public health facilities in India: An analysis of sociodemographic, regional and disease-wise variations

Dear Dr.Yadav,

Thank you for submitting your manuscript to PLOS Global Public Health. After careful consideration, we feel that it has merit but does not fully meet PLOS Global Public Health’s publication criteria as it currently stands. Therefore, we invite you to submit a revised version of the manuscript that addresses the points raised by the reviewers/ 

We look forward to receiving your revised manuscript.

Kind regards,

Roopa Shivashankar, MD, MSc

Academic Editor

Journal Requirements:

1. We do not publish any copyright or trademark symbols that usually accompany proprietary names, eg (R), (C), or TM  (e.g. next to drug or reagent names). Therefore please remove all instances of trademark/copyright symbols throughout the text, including on Table 8.

2. Please update the completed 'Competing Interests' statement, including any COIs declared by your co-authors. If you have no competing interests to declare, please state "The authors have declared that no competing interests exist". Otherwise please declare all competing interests beginning with the statement "I have read the journal's policy and the authors of this manuscript have the following competing interests:"

3. Please provide us with a direct link to the base layer of the map used in Figure 1 and ensure this location is also included in the figure legend. 

Please note that, because all PLOS articles are published under a CC BY license (creativecommons.org/licenses/by/4.0/), we cannot publish proprietary maps such as Google Maps, Mapquest or other copyrighted maps. If your map was obtained from a copyrighted source please amend the figure so that the base map used is from an openly available source.

Please note that only the following CC BY licences are compatible with PLOS licence: CC BY 4.0, CC BY 2.0  and CC BY 3.0, meanwhile such licences as CC BY-ND 3.0 and others are not compatible due to additional restrictions. If you are unsure whether you can use a map or not, please do reach out and we will be able to help you. 

The following websites are good examples of where you can source open access or public domain maps:

Reviewers' comments:

Reviewer's Responses to Questions

**Comments to the Author**

1. Does this manuscript meet PLOS Global Public Health’s publication criteria? Is the manuscript technically sound, and do the data support the conclusions? The manuscript must describe methodologically and ethically rigorous research with conclusions that are appropriately drawn based on the data presented.

Reviewer #1: Yes

Reviewer #2: Partly

Reviewer #3: Yes

2. Has the statistical analysis been performed appropriately and rigorously?

Reviewer #1: Yes

Reviewer #2: Yes

Reviewer #3: Yes

3. Have the authors made all data underlying the findings in their manuscript fully available (please refer to the Data Availability Statement at the start of the manuscript PDF file)?

Reviewer #1: Yes

Reviewer #2: Yes

Reviewer #3: Yes

4. Is the manuscript presented in an intelligible fashion and written in standard English?

Reviewer #1: Yes

Reviewer #2: Yes

Reviewer #3: Yes

5. Review Comments to the Author

Reviewer #1: The authors tried to address the research gaps in unmet need for treatment-seeking and medical consultation, especially for public health facilities across sociodemographic characteristics, regions, and specific diseases. However there are some statistical concerns in this manuscript especially in the methodology, analytical approach and reporting of finding and are described below.

Query 1: In order to address it, authors used data from three rounds of the National Sample Survey (2004, 2014, 2018). However the rationale for performing such comparing the 3 timepoint data analysis is missing, although the population structure, sampling method or sample size differs.

Query 2: In the abstract as well as in main manuscript authors mentioned about “multivariate” analysis. None of the methods adopted in the manuscript is multivariate. It may be corrected as “multivariable analysis”

Query 3: In the “Data and Methods” section, the authors specified that a cross-sectional secondary data from three waves of the National Sample Survey . Whether the sampling strategy remained same for all 3 waves?

Query 4: Page 10, Para 2: Statistical analysis section: Needs to be polished. Authors specified that “In the first part of the analysis, descriptive analysis …….., while in the second part, bivariate estimates … and In the third part of the analysis, …” Instead authors may be instructed to specify the statistic used and test of significance used for bivariate analysis and so on.

Query 5: Page 10, Para 2: Further multinomial logit regression is not a multivariate analysis. It may be modified as multivariable analysis

Query 6: Page 10, Para 2: Moreover The equation for multinomial logistic regression specified is not a general form and is a special case with only two independent variables. This may be replaced with a general form of ‘n’ independent variables

Query 7: Page 10, Last Para : The author mentioned that the analysis was carried out after adjusting the survey design and the sampling weight. Are the weight used same for all the waves or for each wave separate weight computation was done and considered for this analysis? Mention about this in the manuscript for more clarity.

Query 8: Page 11, Table 1: Characteristics described in the table include various socio-demographic characteristics. Hence socio-economic characteristics may be replaced with socio-demographic characteristics and should reflect where ever applicable.

Moreover the sequence of the information differs from the standard representation. Specify the n first followed by the % information. Further, as the sample size varies between the waves, it’s good to provide those number under each columns of NSS-2004, NSS-2014 & NSS-2018.

Query 9: In the titles of Table 1 to 5 mentioned “NSS, 2014-2018” , though 2004 summary were mentioned. This needs to be corrected.

Query 10: Page 15, Table 3: Provide the description of notations and abbreviations used as footnotes

Query 11: Page 16, Table 5: Provide the description of notations and abbreviations used as footnotes

Query 12: Page 19, Table 7: There is some typo error in the title and may be corrected

Query 13: Number of tables found too many, Some of these tables may be kept in the manuscript as supplementary files.

Query 14: Are there any information on Unmet need for treatment-seeking available in NFHS -4 finding. In that case a comparison of these estimates with respect to the findings of NFHS may be described in discussion section.

Query 15: The pattern of trend seen on Unmet need for treatment seeking an well as medical consultation across the different waves may be discussed.

Reviewer #2: The article talks about important topic unmet need of medical consultation in government health care system.

1. The article is written well but for Indian audience. For international reader it has to elaborate in details every terminology used. The manuscript is not written clearly enough to be accessible to non-specialists.

2. The study is secondary data analysis up to year 2018. During COVID Pandemic the health care situation across the country has dramatically changed. Can this study results generalized in current situation? Explain

Title

3. Title is “Unmet need for treatment-seeking” The article mentions more of unmet need of medical consultation. Need to be revise

Abstract

4. “Especially” in abstract: Manuscript does not give details for private health facilities” need to remove this word

Introduction

5. It is written taking public health facilities as a single unit. Assessing Government Medical college, district hospital down to subcentre in single note is lacking clarity for the Indian Health Care System. Need to revised by level of health care for a clear view of international readers.

6. It elaborated only the lacunas and challenges from the health care facility point. Treatment seeking is missing and need to added.

Methodology

7. The term “unmet need” to be standardize across the article. It is bemusing

8. In title, abstract and result it mentions Unmet need for treatment-seeking, unmet need of medical consultation

9. In methodology operational definition are for unmet need for public health facility and unmet need for health care

10. In methodology it also mentions unmet need for medical treatment advised

11. Provide operational definition for Unmet need for treatment-seeking, unmet need of medical consultation, unmet need for medical treatment advised as they same as unmet need for public health facility and unmet need for health care need to be clarified.

12. Details of the methodology is not sufficient to allow to be reproduced

13. Methodology of NSS Survey is written, please revise the methodology to the present study methodology: how the data was retrieved, how were the variable chosen, how much is missing data etc, May be a flowchart

14. Mention what is MPCE Quintile

15. Mention what is SC/ST and OBC

16. Mention about Indian Geography to make readers understand what is North, West, East, Northeast, South, central

Results

17. Authors mentioned “The unmet need for medical consultation at public health facilities remained high at 60% in 2004 to 62% in 2018” Can provide stratification

18. Result is presented monotonously in table format. Table 2 to be in single Map

19. Table 4 Morbidities to be coded in standard format like ICD or SNOMED CT

Discussion

20. The data and analyses do not fully support the conclusion

21. It mentions “One of the important findings of this study is that the main reason for not seeking medical treatment was the ailment not being considered serious enough”. Data description for not seeking treatment is missing in result section.

22. It mentions “This study also shows that the utilization of private healthcare facilities is more common among the elderly population (60+).” Data description private health facility or proportion of government health facility care available is missing in results

Conclusion

23. It mentions “Universal Health Coverage”

24. Compare results with universal health coverage issues in discussion

25. The discussion, result and conclusion need to be synchronized

Minor Changes

26. Page 4 Key Massages replace to Key Messages

27. Keywords to be as MeSH PubMed

28. Table 3 title font size “sociodemographic”

29. In result tables; Non-SC/ST/OBC replace to “unreserved or general”

30. The study itself show sufficient potential that the authors should be encouraged to resubmit a revised version

31. Original data of NSS is used link presented

Reviewer #3: The authors have addressed have a very important topic. Health care seeking and addressing unmet needs of the population in the long run is very crucial for policy makers to achieve universal health coverage in the country. I have few suggestions and comments:

1. A separate annexure may be provided to mention which block of NSSO data was used and how the data was merged. I am concerned whether the variables/questions remained the same across the 3 rounds and the answers/choices provided by the respondents had the same coding.

2. More details of multi-collinearity screening need to be added. May be in the annexure.

3. A chi-square for trend can be added in Table 1.

4. Can Table 2 be shown as a figure across 6 regions (separately for treatment seeking and consultation from public facility)??. The table appears too crowded. On seeing a trend figure, I am sure more interesting points can come out.

5. Elaborate and provide the full form of IHDS-II. Is this similar to NSSO? Can be mentioned in the methods too.

6. In Table 3 can we have the ‘p’ values. That will be more useful. Remove the “***”

7. Why Table 4 doesn’t have a Chi-square done with a ‘p’ value? I would like to see the change over survey rounds and not within rounds. This applies to all tables.

8. The *, # and ns can be removed in Table 5. The confidence intervals are more than enough to interpret whether the aOR was significant or not.

9. Table 6. I recommend for uniformity, it can be aligned similar to Table 3 & Table 4. Take Quality and show the values for 2004, 2014 and 2018 side by side. Take waiting and show the values of 2004, 2014 and 2018 side by side. Show the chi-square for trend ‘p’ value.

10. Table 7. Same as above

11. Consent for publication needs to be changed to PloS Global health

12. In the author contribution, I noticed a RR. Are we missing anyone or is it RY?

13. The Table S2 can mention which blocks provided which variable and how merging was done separately at the individual, household and community level. Change education of the youth to education of the patients.

6. PLOS authors have the option to publish the peer review history of their article (what does this mean?). If published, this will include your full peer review and any attached files.

**Do you want your identity to be public for this peer review?** For information about this choice, including consent withdrawal, please see our Privacy Policy.

Reviewer #1: No

Reviewer #2: No

Reviewer #3: **Yes: **Dr Giridara Gopal Parameswaran

---

## [Decision Letter · Decision Letter 1]

28 Jan 2022

PGPH-D-21-00530R1

Unmet need for treatment-seeking from public health facilities in India: An analysis of sociodemographic, regional and disease-wise variations

Dear Dr. Yadav,

Thank you for submitting your manuscript to PLOS Global Public Health. After careful consideration, we feel that it has merit but does not fully meet PLOS Global Public Health’s publication criteria as it currently stands. Therefore, we invite you to submit a revised version of the manuscript that addresses the points raised during the review process.

Please address the minor comments by the reviewers. Ensure the manuscript is proofread for English language and typos. 

We look forward to receiving your revised manuscript.

Kind regards,

Roopa Shivashankar, MD, MSc

Academic Editor

Journal Requirements:

Additional Editor Comments (if provided):

Reviewers' comments:

Reviewer's Responses to Questions

**Comments to the Author**

1. If the authors have adequately addressed your comments raised in a previous round of review and you feel that this manuscript is now acceptable for publication, you may indicate that here to bypass the “Comments to the Author” section, enter your conflict of interest statement in the “Confidential to Editor” section, and submit your "Accept" recommendation.

Reviewer #1: All comments have been addressed

Reviewer #2: All comments have been addressed

Reviewer #3: All comments have been addressed

2. Does this manuscript meet PLOS Global Public Health’s publication criteria? Is the manuscript technically sound, and do the data support the conclusions? The manuscript must describe methodologically and ethically rigorous research with conclusions that are appropriately drawn based on the data presented.

Reviewer #1: Yes

Reviewer #2: Yes

Reviewer #3: Yes

3. Has the statistical analysis been performed appropriately and rigorously?

Reviewer #1: Yes

Reviewer #2: Yes

Reviewer #3: Yes

4. Have the authors made all data underlying the findings in their manuscript fully available (please refer to the Data Availability Statement at the start of the manuscript PDF file)?

Reviewer #1: Yes

Reviewer #2: Yes

Reviewer #3: Yes

5. Is the manuscript presented in an intelligible fashion and written in standard English?

Reviewer #1: Yes

Reviewer #2: No

Reviewer #3: Yes

6. Review Comments to the Author

Reviewer #1: (No Response)

Reviewer #2: Affiliation: PHD to Ph.D.

Abstract: Result- last line "unment" correct the spelling

Rationale :

national rural health mission : Uppercase

Outcome variables:

first para Second last line "logtistic and multinomial" correct the spelling

last line " The operational definitions of all the these outcomes variables are

given below" to be shifted to appropriate place

third para- first line "three-tyre health system" to three-tier

last word "thospitals" correct the spelling

Reason for not seeking treatment at all from any sources: "neibourhood" correct the spelling

Reasons for unmet need for treatment seeking any public health facilities:"categoried them into" correct the spelling

Statistical analysis:

"treatrment" correct the spelling

last para "three rounds sepretaly" correct the spelling

Tables foot notes: Sheduled Caste/Sheduled Tribes correct the spelling

OBC=Others Backwords Classes, correct the spelling

Figure 3 Caption " unmet" uppercase

Reviewer #3: The authors have built the background to the analysis and the rationale very well this time around.

7. PLOS authors have the option to publish the peer review history of their article (what does this mean?). If published, this will include your full peer review and any attached files.

**Do you want your identity to be public for this peer review?** For information about this choice, including consent withdrawal, please see our Privacy Policy.

Reviewer #1: **Yes: **Binukumar Bhaskarapillai

Reviewer #2: No

Reviewer #3: No

---

## [Editor Report · Decision Letter 2]

17 Feb 2022

Unmet need for treatment-seeking from public health facilities in India: An analysis of sociodemographic, regional and disease-wise variations

PGPH-D-21-00530R2

Dear Dr. Yadav,

We are pleased to inform you that your manuscript 'Unmet need for treatment-seeking from public health facilities in India: An analysis of sociodemographic, regional and disease-wise variations' has been provisionally accepted for publication in PLOS Global Public Health.

Best regards,

Roopa Shivashankar, MD, MSc

Academic Editor